# A Review on the Use of Self-Curing Agents and Its Mechanism in High-Performance Cementitious Materials

**Norhaliza Hamzah** [1,2,*], **Hamidah Mohd Saman** [1], **Mohammad Hajmohammadian Baghban** [3,*], **Abdul Rahman Mohd Sam** [2], **Iman Faridmehr** [4], **Muhd Norhasri Muhd Sidek** [1], **Omrane Benjeddou** [5] and **Ghasan Fahim Huseien** [6,*]

1 School of Civil Engineering, College of Engineering, Universiti Teknologi MARA, Shah Alam 40450, Malaysia; hamid929@uitm.edu.my (H.M.S.); norhasri@gmail.com (M.N.M.S.)

2 UTM Construction Research Center, Universiti Teknologi Malaysia, Skudai 81310, Malaysia; abdrahman@utm.my

3 Department of Manufacturing and Civil Engineering, Norwegian University of Science and Technology (NTNU), 2815 Gjøvik, Norway

4 Institute of Architecture and Construction, South Ural State University, Lenin Prospect 76, 454080 Chelyabinsk, Russia; s.k.k-co@live.com

5 Department of Civil Engineering, College of Engineering, Prince Sattam Bin Abdulaziz University, Alkharj 16273, Saudi Arabia; benjeddou.omrane@gmail.com

6 Department of the Built Environment, School of Design and Environment, National University of Singapore, Singapore 117566, Singapore

* Correspondence: norhalizahamzah@utm.my (N.H.); mohammad.baghban@ntnu.no (M.H.B.); bdggfh@nus.edu.sg (G.F.H.)

**Abstract:** Self-cured concrete is a type of cement-based material that has the unique ability to mitigate the loss rate of water and increase the capacity of concrete to retain water compared to conventional concrete. The technique allows a water-filled internal curing agent to be added to the concrete mixture and then slowly releases water during the hydration process. Many researchers have studied the composition of self-curing concrete using different materials such as artificial lightweight aggregate (LWA), porous superfine powders, superabsorbent polymers (SAP), polyethylene glycol (PEG), natural fibers, and artificial normal-weight aggregate (ANWA) as curing agents. Likewise, physical, mechanical, and microstructure properties, including the mechanisms of curing agents toward self-curing cement-based, were discussed. It was suggested that adopting self-curing agents in concrete has a beneficial effect on hydration, improving the mechanical properties, durability, cracking susceptibility behavior, and mitigating autogenous and drying shrinkage. The interfacial transition zone (ITZ) between the curing agent and the cement paste matrix also improved, and the permeability is reduced.

**Keywords:** high-performance concrete; self-curing; curing agents; mechanism; interfacial transition zone

## 1. Introduction

Curing is a process of maintaining the rate and the extent of moisture loss within a proper temperature in concrete during cement hydration and reduces water evaporation [1–4]. Curing allows continuous hydration of cement until achieving its potential strength and durability. However, it is critical to ensure that the moisture condition is appropriate; otherwise, the hydration of cement virtually ceases due to the relative humidity within the capillaries falling below 80% [1,5,6]. If hydration ceases, sufficient calcium silicate hydrate (C-S-H) cannot be developed [7,8], which disrupts the development of dense microstructure and the refined pore structure within the cement matrices allowing the ingress of deleterious agents into the concrete. These subsequently lead to poor quality

of concrete, such as causing plastic shrinkage cracks, poorly formed hydrated products, finishing issues, and other surface defects [4,9,10].

Previous studies have reported various methods to prevent moisture loss during concreting like spraying or fogging the surface of newly cast concrete with water [1,11–14], applying wet surface covering [3,13,14], and water ponding which is suitable for horizontal surfaces [13–15]. Other methods mentioned include membrane curing, which retains the water within the concrete to maximize the potential hydration [3,10,13,14], steam [11,16–18], and leaving formwork in place [1,15]. Nevertheless, the accelerated curing methods such as microwave curing [19–21], direct electric curing [22–24], and infrared curing [25] are used in the application of heat on fresh concrete to promote rapid cement hydration by securing early-age strength of concrete. However, a more common feature of all existing curing techniques is frequently applied to the surface of the concrete. If the capillary porosity in the concrete is disconnected during the curing process, moisture is unable to penetrate the entire depth of the concrete, limiting the effectiveness of the curing process.

Therefore, a new approach of self-curing concrete has been introduced [26,27]. Self-curing or internal curing concrete can be defined as cement-based material having additional water capacity during the curing regime for the hydration process [28]. The practice of self-curing is a feasible technique that can supply more water to concrete towards more effective cement hydration and decrease self-desiccation. The following sections of this study present a review of the techniques which have been studied in self-curing agent reported by researchers, including its mechanism.

## 2. Mechanism of Hydration of Conventional Concrete

The addition of water to ordinary Portland cement (OPC) powder initiates the cement hydration reactions instantly. This series of chemical reactions leads in the cement paste setting and hardening. Within a few minutes, needle-like crystals of calcium sulfoaluminate hydrate, notably ettringite, develop. After a period of time, ettringite converts into monosulfate hydrate [22]. Two hours after the cementation process begins, large prismatic crystals of calcium hydroxide (CH) and tiny calcium silicate hydrates (C–S–H) fill the voids formerly filled by water and hydrated cement particles (as shown in Equations (1)–(6)). Therefore, calcium silicate hydrate, calcium hydroxide, and calcium sulfoaluminate are the three primary components of hydrated cement paste. Calcium silicate hydrate is the primary hydration product, contributing to almost 60% of the volume of solids. It is composed of a layer of sponge-like structures with a huge surface area (500 $m^2$/g). The ultimate strength of the product is largely owing to the development of C-S-H and is principally due to van der Waals physical adhesion forces. Calcium hydroxide is the second most prevalent component, contributing to approximately 25% of the total. Compared to C–S–H, it is composed of massive plate-like crystals with a lower surface area [22,25]. It contributes to the reduction of van der Waal forces and is relatively soluble in comparison to C–S–H, making the concrete reactive to acidic solutions. Calcium sulfoaluminate plays a small part in the cementitious structure properties by almost 15% solid volume. Chemical resistance of the cementitious final product to sulfate attack is an issue, owing to the existence of the monosulfate hydrate. Figure 1 shows electron microscopic images of hardened cement paste after hydration.

$$2\,Ca_3SiO_5 + 7\,H_2O \ \leftrightarrow\ Ca_3Si_2O_3(OH)_8 + 3\,Ca(OH)_2 \tag{1}$$

$$2\,Ca_2SiO_4 + 5\,H_2O \ \leftrightarrow\ Ca_3Si_2O_3(OH)_8 + \ Ca(OH)_2 \tag{2}$$

$$Ca_4Al_2Fe_2O_{10} + 7\,H_2O \ \leftrightarrow\ Fe_2O_3 + \ Ca_3Al_2(OH)_{12} + \ Ca(OH)_2 \tag{3}$$

$$CaO \ + \ H_2O \ \leftrightarrow\ Ca(OH)_2 \tag{4}$$

$$Ca_3Al_2O_6 + 3\,CaSO_4 \cdot 2H_2O \ + 26\,H_2O \ \leftrightarrow\ [Ca_3Al(OH)_6 \cdot 12H_2O]_2(SO_4) \cdot 32H_2O \tag{5}$$

$$[Ca_3Al(OH)_6 \cdot 12H_2O]_2 \cdot (SO_4)_3 \cdot 2H_2O \ + \ Ca_3Al_2O_6 + 4\,H_2O \ \leftrightarrow \\ 3\,[Ca_2Al(OH)_6 \cdot 2H_2O]_2SO_4 \cdot 2H_2O \tag{6}$$

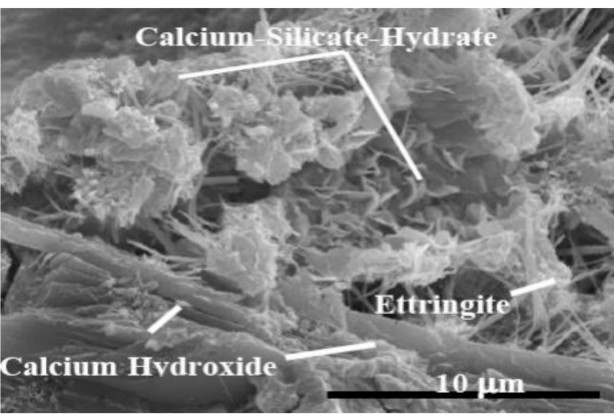

**Figure 1.** Electron microscopic images of hardened cement paste after hydration [13].

### 3. Self-Curing Agent and Mechanism in Cementitious Materials

Self-curing technology or internal curing has gained popularity within the concrete community research field. The concept of self-curing concrete or mortar is to reduce the evaporation of water in concrete and improve the water retention capacity in concrete [27–30]. As a result, the technique has been readily introduced where water-filled internal curing agents acting as reservoirs are added to the concrete mixture which will gradually release water during hydration and evaporation process [31–34], as illustrated in Figure 2. High-performance concrete (HPC) mixtures were initially developed given the growing issues regarding concrete durability [35,36] and due to the use of lower water-cementitious (w/c) material ratios, in addition to chemical admixtures and supplementary cementitious material (SCMs). Measuring the extent of hydration in the cementitious system is a key indicator that leads towards achieving the good performance of the concrete [37]. Low w/c ratio concrete mixtures, less than 0.42, are unable to fully hydrate the cement in the mixture due to insufficient water [1]. Therefore, the benefit of self-curing concrete from absorbed moisture in porous aggregate was discovered to solve the problem relating to insufficient water in HPC mixtures by providing extra water to replace that which was depleted during the process of cement hydration.

Many researchers have investigated self-curing concrete composition using different materials as curing agents, such as porous aggregate, for example lightweight aggregate (LWA) [38–41], porous superfine powders [42–45], artificial normal-weight aggregate [46–48], chemical curing agents, for example superabsorbent polymers [49–53] and polyethylene glycol [54–56], and natural fibers [33,57,58]. The following sub-sections explain the mechanism and properties of self-curing concrete using different curing agents.

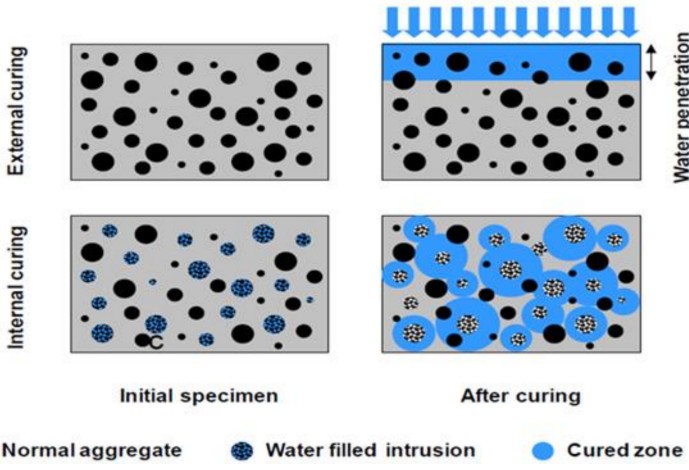

**Figure 2.** Illustration of the differences between self-curing and external curing [59].

Porous aggregate is frequently associated with poor concrete quality. However, when utilized in wet conditions, the aggregate might benefit the concrete since the water absorbed by the aggregate is slowly released into the already-hardened cement paste, continuing the hydration process. As a result, concrete properties such as increased strength and decreased drying shrinkage will be improved. The water movement is caused by the humidity gradient between the aggregate that is high and the cement paste that is low. According to [28] schematic diagram of the mechanism of self-curing concrete depicted as Figure 3. Self-curing, also known as autogenous curing or internal curing, enables curing "from the inside out", which is achieved by introducing a pre-saturated component as an internal curing agent. The curing agent is spread uniformly throughout the matrix and acts as a reservoir for internal water. The water within the curing agent has not involved in the chemical reaction until a humidity gradient forms during an initial hydration phase. On [29], it was illustrated that the self-curing process occurs as shown in Figure 4. Water is transported from the curing agent to un-hydrated cement by the driving forces of capillary suction, vapor diffusion, and capillary condensation for supporting continuous hydration. As result, chemical shrinkage and self-desiccation due to low w/b can be significantly reduced.

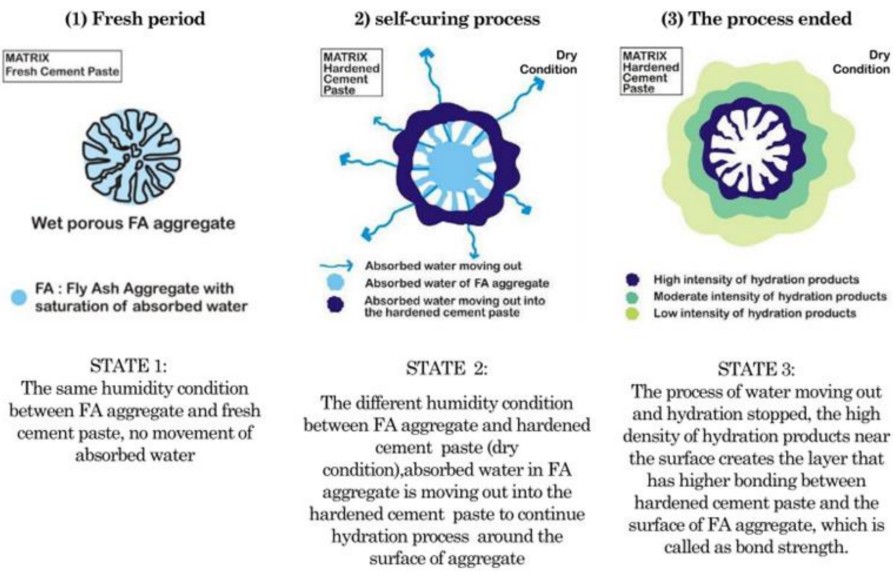

**Figure 3.** The mechanism of self-curing concrete [28].

### 3.1. Artificial Lightweight Aggregate (LWA)

Prewetted lightweight aggregates have often been used as internal reservoirs in which a system of capillary pores in cement paste is formed during hydration, and as soon as the relative humidity (RH) decreases (due to hydration and drying), a humidity gradient develops [35,60–62]. The migration of water in concrete based on the law of fluid flow and the system's law of capillary attraction is illustrated in Figure 5. As observed in the figure, the radius of pores in cement paste (r(t)) is smaller than the pores in LWA (Ra). The pores of the cement paste by capillary suction absorbs the water from the LWA due to difference in vapor pressure and transports the water to the drier cement paste, where a reaction with the un-hydrated cement occurs [35,62–64]. The un-hydrated cement particles, hydrated to form hydration products, reduce the size of the pores, enabling the pores to continue absorbing the water from the LWA. This process continues until all the water from LWA has been transported to the cement paste, creating a self-curing mechanism.

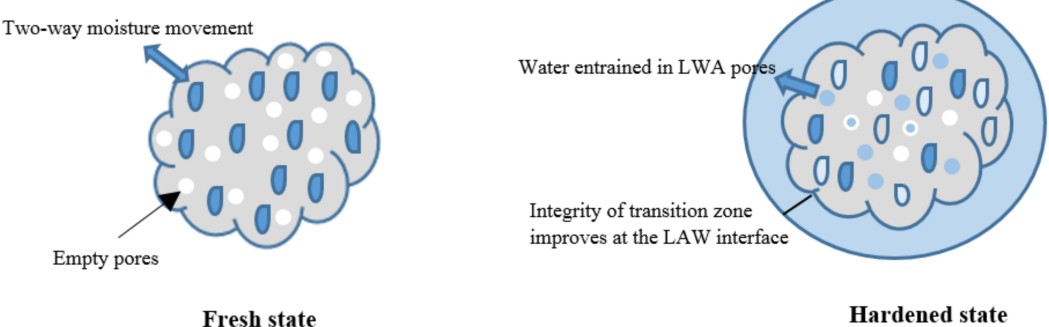

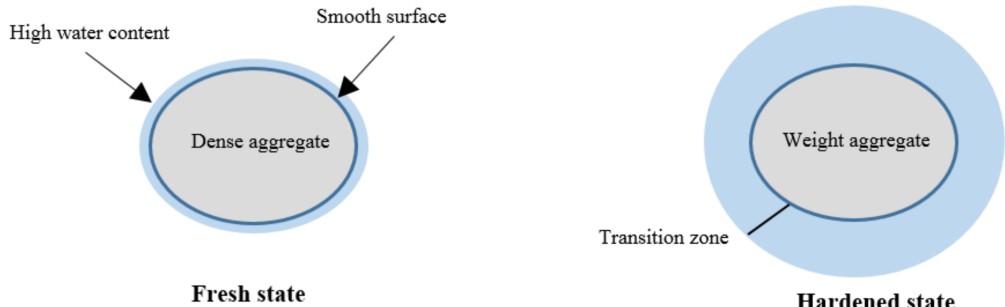

**Figure 4.** The contact zone under internal curing and normal curing.

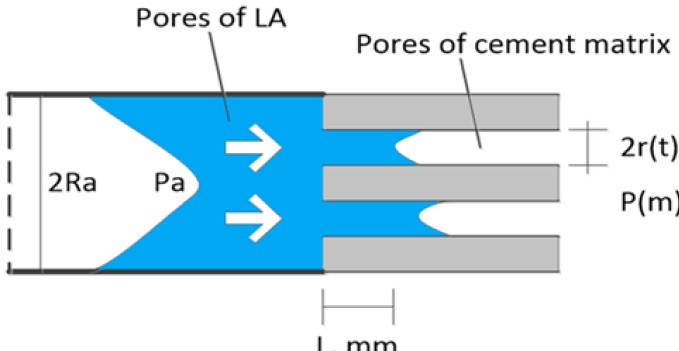

**Figure 5.** Model illustrating the movement of water in cement-based materials, incorporating self-curing with r(t) < Ra [64].

Among the various features found in LWA, it has been shown to have high porosity and absorption capacity with the added benefit of supplying curing water internally [38,65,66]. Moreover, due to contain of porosity, the density of LWA is less than that of artificial NWA. The saturated-surface dried density of LWA is reported to be below 2000 kg/m$^3$ [65,67–71]. However, water absorption of porous LWA depend on the pore structures. The minimum water absorption above 5% by mass of LWA is required as recommended by ASTM C1761/C1761M [72–74]. LWA can be in the form of coarse and fine aggregate and the more common aggregates used as a curing agent in self-curing concrete, which are natural-based, are expanded shale [75,76], expanded clay (LECA), [38,42,65,70,77–80] and pumice [68,81,82]. Whereas the curing agent from by-product materials, often studied by researchers, is bottom ash [4,69,70,83–86].

### 3.2. Porous Superfine Powders

Porous superfine powders are small particles with a large specific surface area and a mesoporous structure. It can absorb the aqueous phase, enabling water supply for the hydration process in the cementitious material [32,87]. Porous superfine powders possess nanometre-size pores, for instance, cenosphere [43,88–90], rice husk ash (RH) [44,45,91,92], and biochar [93,94]. Generally, the particle size of porous superfine powders ranges between 5 μm and 10 μm, which is much smaller compared to SAP particles and LWA, where the pore size ranges between 4 nm and 10 nm. Only if the pore size of cement paste is smaller than superfine powders' pore size will the water be released into the cement paste, in which the hydration process occurs. In addition, the application of porous superfine powders capable of slowing down the internal RH (self-desiccation) in UPHC significantly reduces its autogenous shrinkage [32,44,95]. According to the Kelvin equation [96], the capacity of water saturation would respond to changes in humidity betwixt 75% and 98%, whereas the pore size range corresponds to a change in RH of around 75 to 98%. It is assumed that the water stored in the mesopores will slowly release its water when the internal RH in concrete drops below 98% to compensate for self-desiccation during hydration. By using the concept of protected paste volume [97], it revealed that cement paste should be closed to the internal curing water reservoir so that the absorbed water could be penetrated. Thus, cement paste is protected from self-desiccation by the absorbed water. To achieve this, the curing agent particle size should be as small as possible [32,98].

### 3.3. Artificial Normal Weight Aggregate (ANWA)

Waste material in the construction industry such as ceramic [48,99,100] and recycled concrete waste [101–104] have the potential as water reservoirs in self-curing concrete given their ability to absorb water due to its porosity. Zou et al. [105] mentioned that the SSD density of crushed waste ceramic was 2.48 g/cm$^3$, where the value is almost similar to that of natural normal fine aggregate. This statement is further strengthened by Shigeta et al. [48], who reported that the SSD density of waste ceramic coarse aggregate is 2.26 g/cm$^3$ while natural normal coarse aggregate is 2.62 g/cm$^3$. Thus, it provides a good effect on the strength of concrete containing crushed waste ceramic material compared to that of plain concrete. Moreover, water absorption of waste ceramic coarse aggregate was 9% compared to natural normal aggregate which recorded 0.67% water absorption. While Suzuki et al. [99] revealed that water absorption of waste ceramic coarse aggregate was 9%, and the crushing rate value was 21.4%, almost similar to that reported by Sato et al. [100]. Thus, the capability of water absorption in ceramic waste aggregates will help in the hydration process of the concrete.

### 3.4. Superabsorbent Polymer (SAP)

SAP was initially developed during the 1980s and has since been widely used in forestry, agriculture, health supplies, and in other fields given their potential as a water reservoir and their ability to expand and retain water [106–111]. The capability of SAP has been used with cementitious materials in concrete to mitigate shrinkage (autogenous and drying) via self-curing [107,112–116] to enhance the durability toward freeze and thaw deterioration [110,117,118]. Superabsorbent polymers (SAP) are recognized as hydrogels, consisting of a three-dimensional cross-link network structure that can absorb a large volume of liquid compared to their mass because of osmotic pressure and expand to form an insoluble gel [107,118–120]. A chemical reaction will eventuate when SAP is exposed to an aqueous solution, leading to shrinkage or swelling of the SAP. The absorption of SAP is driven by osmotic pressure, as illustrated in Figure 6, before it develops the space between cross-links and polymer chains. The presence of osmotic pressure originates from a concentration gradient of moveable ions between the gel and solution [110,118,121,122]. The swollen SAPs then react as water reservoirs in the concrete. However, as the humidity in the concrete decreases, the absorbed water is pulled back into the cement paste capillary

pores. This leads to the SAPs gradually releasing the absorbed water and leaving the voids [118,123].

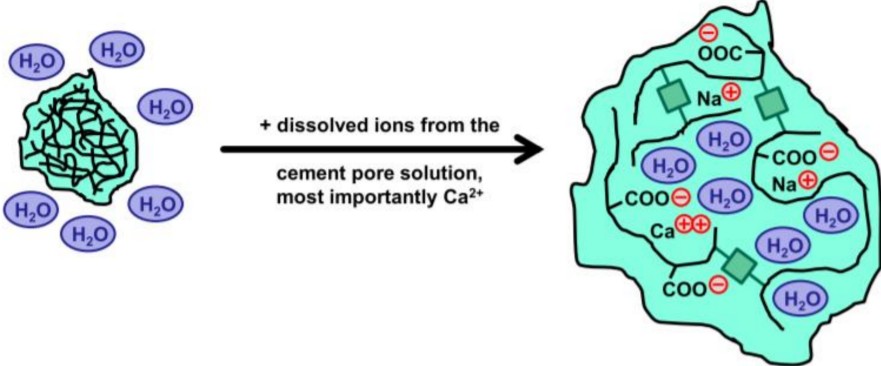

**Figure 6.** Process of water uptake to SAP [110].

### 3.5. Polyethylene Glycol (PEG)

Polyethylene-glycol is a condensing polymer of ethylene oxide and water with general formula $H (OCH_2CH_2) nOH$, where n is the average number of repeated groups of ox ethylene usually between 4 and about 180 [26]. According to Raoult's Law, when the vapor pressure of the solute in the pure condition is less than the vapor pressure of the solvent in the pure condition, it is apparent that theoretically, by adding additives, the vapor pressure of water will decrease, thus reducing the evaporation rate above the concrete surface [26,27,34,55,56]. Therefore, the application of water-soluble polymers for instant PEG as self-curing in concrete has been observed to be both effective and efficient in retaining water and enhancing the hydration process [34,124,125]. Moreover, with water molecules, polymeric chains created by hydrophilic units form hydrogen bonds. A hydrogen bond is a frail bond formed in a compound between hydrogen atoms and strongly electronegative atoms in other molecules [26,126]. The existence of a positive charge at the hydrogen atom causes attraction to the electronegative atom electrostatically as illustrated in Figure 7. To this end, water soluble polymers with either hydroxyl (-OH-) or ether (-O-) functional groups have been used as the chemical to minimize the impact of self-desiccation in concrete [26,126]. Previous researchers have also investigated water retention, hydration, compressive strength, microstructure characteristic, and durability of concrete using PEG as a curing agent [55,56,77,125,127].

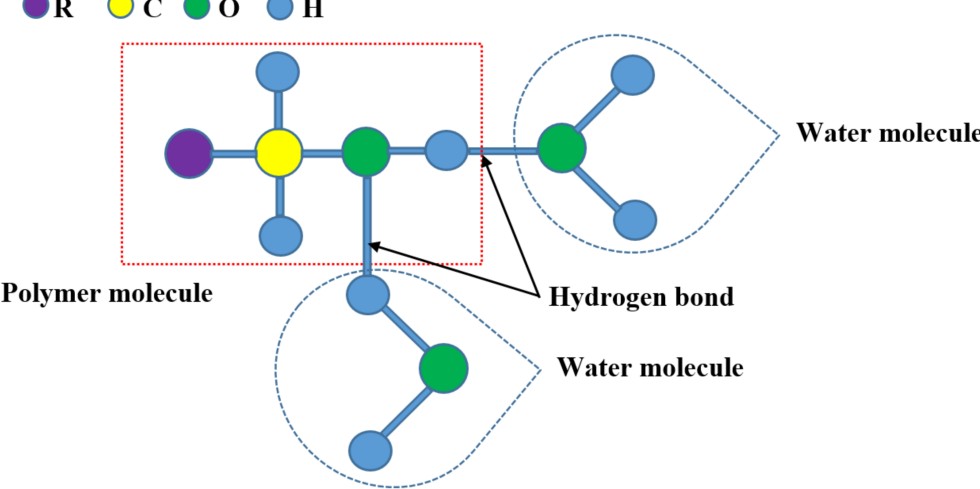

**Figure 7.** Hydrogen bonds between water molecules and an –OH group on a polymer molecule.

### 3.6. Natural Fibers (NF)

Previous researchers investigated that wood-derived fibers and powder have potential as self-curing agents in cement-based material due to the former's capability to absorb and retain water in addition to gradually releasing absorbed water [33,57,58,128–131]. Good examples of wood-derived fibers used as self-curing agents in concrete are eucalyptus pulp [33], kenaf fibers [58], and cellulose fibers [57]. Wood-derived fibers are hygroscopic materials, and the movement of water via the pulps depends on the concentration gradient (diffusion) by capillary draw and the effect of osmotic pressure [33,131,132]. Moaven-zadeh [133], Elsaid et al. [130] and Jongvisuttisun et al. [33] explained that wood-derived fibers consist of two pores, namely larger pores (e.g., lumen) comprised of free water and smaller pores, as illustrated in Figure 8. Both pores play a key role in the transportation of moisture, from the wood pulp to nearby hydrating cement. Furthermore, the pore solution in cement-based material is alkaline, which therefore influences the character of wood pulp to swell or shrink and to change the effective size of the porous space [134–136], thus affecting the water transport.

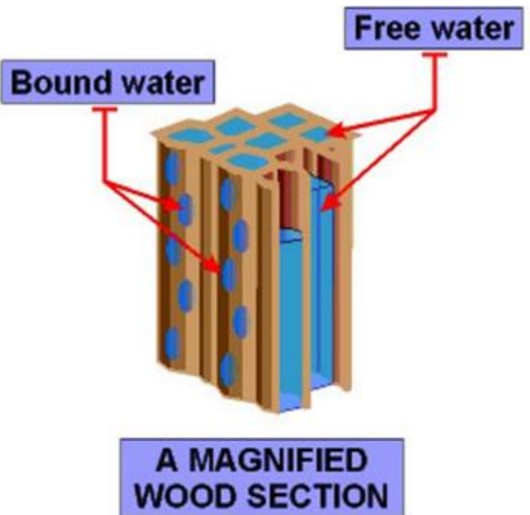

**Figure 8.** Free and bound water in wood [137].

## 4. Effects of Self-Curing Agents on Properties of Concrete or Mortar

### 4.1. Workability

The impact of porous aggregate on the workability of concrete is rarely discussed. The explanation might be that soaked-porous aggregate neither absorbs nor releases water before setting. Hence its workability is unaffected. However, if the dry porous aggregate is added to the mixed concrete, the absorption rate will be slower, resulting in bleeding and segregation of the mixture at the first stage [138]. Thus, a few minutes of pre-mixing of porous aggregate and water is also suggested [139]. In addition, spherical aggregates have been found to improve the workability of fresh concrete [140–143] due to having smaller intrinsic viscosity than other shapes.

Studies have also shown that the addition of SAP caused workability reduction and delays the setting time of concrete [111,144]. Nevertheless, the prewetted SAP resulted in an increasing slump when SAP volume increased. It shows that the spherical particles that pre-absorbed SAP might act as lubricant in concrete mixture, reducing friction between paste and aggregate [111].

### 4.2. Compressive Strength

Several researchers discovered that porous aggregate decreases the strength of high-performance concrete and the strength stays decreased when the replacement of prewetted porous aggregate is increased [76,145,146]. Other researchers revealed that the reduction

in concrete strength is due to the low strength of porous aggregate itself [41,138,147,148]. The detrimental impact of porous aggregate on concrete strength can be mitigated by reducing the size of porous aggregate and improving its distribution [43,93,149]. Generally, the results of compressive strength of concrete at earlier and later ages increased if the optimum proportion was obtained, usually about 20% to 40% replacement of conventional aggregate [104]. However, several researchers revealed that the compressive strength of concrete decreased if the replacement of porous aggregate was more than 50% [38,69,104,150], as shown in Figure 9. The additional water supplied by the prewetted porous aggregate promotes a higher degree of hydration. It fills the pores with hydrated products (C-S-H gel), resulting in an improvement in compressive strength in concrete [63,151]. Agostini et al. [152] discovered that porous aggregate reduced the amount of CH while increasing the density of C-S-H at the interface between the aggregate and the cement paste.

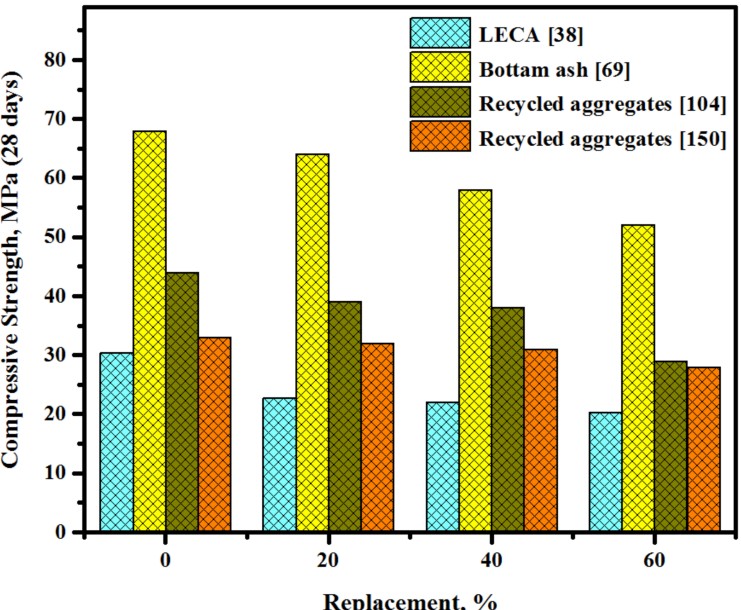

**Figure 9.** Compressive strength of porous aggregate with percent replacement of conventional aggregate at 28 days.

Many researchers reported SAP improving the compressive strength [32,36,42,80,107,153] of high-performance concrete. However, some of the compressive strength decreased as greater amounts of SAP were added to the concrete [53,154–158], as shown in Table 1. Song et al. [154] studied the effect of SAP on the compressive strength of concrete and indicate that the decreasing of strength due to SAP augments in the concrete. SAP swelled after absorbing water, becoming hydrogels and acting as voids in the cementitious materials [159], as depicted in Figure 10. The results also revealed that the ratio of early-age strength reduction due to SAP addition was more pronounced in the concrete specimen than in the concrete specimen without SAP. Nevertheless, the ratio of the later-age strength depended on the SAP dosage used in the specimen (Table 1). Strength development was increased due to the self-curing by SAP, which resulted in enhanced hydration in the specimens. These results are comparable to observations in other studies [160,161].

**Table 1.** Compressive strength of mixtures with SAP at 28 days.

| Researchers | w/c | Compressive Strength, MPa | | | |
| --- | --- | --- | --- | --- | --- |
| | | Reference Concrete | Cement-Based with SAP Addition (% by Weight of Binder) | | |
| [158] | | | 0.1 | 0.2 | 0.3 | 0.8 |
| | 0.3 | 49 | 52 | 44 | 39 | 34 |
| [155] | | | 0.05 | 0.16 | 0.26 | - |
| | 0.33 | 64 | 62.5 | 59.4 | 57.3 | - |
| [53] | | | 0.05 | 0.09 | 0.14 | - |
| | 0.3 | 67 | 73 | 72 | 61 | - |
| [156] | | | 0.2 | 0.4 | 0.6 | - |
| | 0.3 | 120 | 114 | 105 | 100 | - |
| [157] | | | 0.57 | 0.86 | 1.14 | - |
| | 0.3 | 66.71 | 62.26 | 58.16 | 49.28 | - |
| [144] | | | 0.3 | 0.6 | - | - |
| | 0.3 | 107 | 99 | 93 | - | - |
| [154] | | | 0.15 | 0.3 | - | - |
| | 0.4 | 45 | 35 | 25 | - | - |

The inclusion of water-soluble polymer self-curing agent, polyethylene glycol (PEG) has significantly resulted in an increase of compressive strength in cementitious material compared to specimen without PEG, as reported by previous researchers [162–164]. Mousa et al. [163] studied mixes with and without PEG, prepared and cured in laboratory air at 25 °C. They founded that the samples containing 2% of self-curing agent show 32.5% compressive strength increase at age of 28 days compared to the samples without self-curing agent. The pronounced effect on the compressive strength might be due to normal concrete mixes that were air cured. Vaisakh et al. [164] found that compressive strength air cured mixes with a PEG increase of 5.41% compared to water cured normal mixes. These reports are similar to the study by Rizzuto et al. [162].

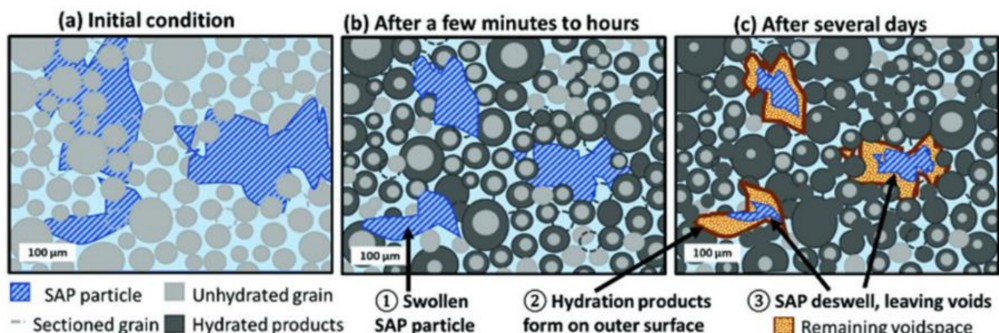

**Figure 10.** (**a**) Soaked SAPs in the cement mixture; (**b**) SAPs gradually release the absorbed water; (**c**) SAPs de-swell and leave voids [165].

### 4.3. Autogenous Shrinkage and Drying Shrinkage

Prewetted porous aggregate can more effectively minimize autogenous shrinkage compared to the oven-dry porous aggregate [80,166–169]. This is because of the dry porous aggregate's inability to absorb sufficient additional water from concrete during the plastic stage [139]. Dry porous aggregate can achieve almost the same result as prewetted porous aggregate if it can absorb the required amount of extra water from the concrete mixture [170].

Prewetted porous aggregate gradually released internal curing water, extending the time that internal RH remains at 100%. As a result, this reduces autogenous shrinkage whereby refining pore structure and improving the hydration [138]. Besides, the kinds of LWA might have an impact on their efficacy. Zhuang et al. [171] discovered that prewetted porous aggregate with higher water absorption and lower strength reduces autogenous shrinkage. However, porous aggregate with higher strength and lower water absorption will have a better effect if put in the dry condition. In general, porous aggregate minimizes the total shrinkage at 28 days [172], while nevertheless increasing drying shrinkage [146]. According to Costa et al. [146], a 34% increase in drying shrinkage was observed in the specimen consisting of fine pumice aggregate compared to the reference specimen with conventional aggregates. The increase in drying shrinkage is related to lower elastic modulus and increased water binder ratio [173,174].

The development rate in autogenous shrinkage of high-performance concrete can be minimized by adding SAP (Table 2). Thus, the emergence of cracking can be delayed [175]. The dosage [153,175,176], type [52,56,107,153], particle size [153,175,176], and water-saturated state of SAP [177] play an important role in the efficiency of self-curing. Shen et al. [155] reported that the autogenous shrinkage of self-cured concrete decreased with increase of internal curing water provided by SAP. The rate of autogenous shrinkage reduced as the concrete ages increased, while the autogenous shrinkage rates of all mixtures were remarkably similar at 28 days. The effect of self-curing with varied dosages of SAP on autogenous shrinkage can be related to the extra water supplied by SAP [113]. The water inside the SAP acts as the water contained inside pores at the time of batching, and it is available to assist internal curing while not affecting the initial w/c [178,179]. Internal relative humidity decreases as a result of water loss by self-desiccation during the hydration process, resulting in driving force [107]. Thus, internal curing water is released from SAP into the cement paste, contributing to continuing hydration of cement. A higher dosage of SAP depicts more water being released when autogenous shrinkage is reduced. The drying shrinkage also reduced with the increase of dosage of SAP as reported by Jensen and Hansen [153], Ma et al. [138], Assmann and Reinhardt [180], and Kong et al. [181]. However, if the environment is arid, the water from the SAP particles evaporates quickly, resulting in cracking. The additional water absorbed by the self-curing agent has a considerable impact on the drying shrinkage of self-cured cement-based material.

Using a chemical curing agent such as PEG was also reported to reduce early age shrinkage cracks [182–184]. Amin et al. [183] investigated engineering properties on self-curing concrete by using polyethylene glycol. The results revealed that adding PEG to the concrete mixes to perform the self-curing role contributes to reducing dry shrinkage compared with reference concrete. Hence, the application of the self-curing explained in this research increased the subsequent degree of hydration of cement and the chemical shrinkage, consequently effectively reducing early and late shrinkage. However, only a few researchers had investigated the autogenous and drying shrinkage of concrete when PEG was used as a curing agent.

### 4.4. Effect of Curing Agent on Interfacial Transition Zone

The interfacial transition zone (ITZ) between cement pastes and aggregate is the most important interface in concrete. There is high porosity and more calcium hydroxide ($Ca (OH)_2$) and ettringite in ITZ between cement paste and conventional aggregate [141]. However, a thin and denser ITZ was formed between porous aggregate and cement paste matrix [168,185–187], as depicted in Figure 11.

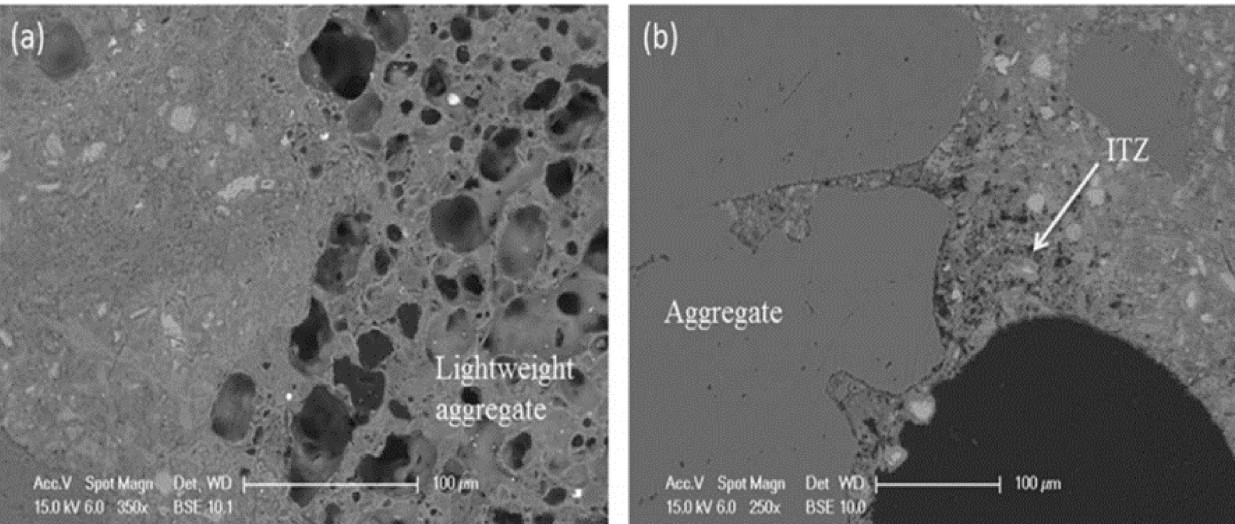

**Figure 11.** ITZ microstructure in cement mortar (**a**) with porous aggregate, (**b**) with conventional aggregate [187].

The formation of C-S-H in high quantities [42,99,104,152] and the homogeneousness [188,189] are also improved in ITZ. Thus, it enhances the strength of internally curing concrete and establishes a well bonding [104] between porous aggregate and cement paste at ITZ. Moreover, some hydrated products that were discovered in pores of porous aggregate contributed to an increase in strength at later ages [42,190,191]. Sun et al. [187] applied SEM techniques to compute the ITZ microstructure, founded a 3D digital model of ITZ using 3D image reconstruction techniques, and applied the mesoscale chemo thermal-hydraulic model to simulate the development of ITZ. The research revealed that internal curing can improve the durability of concrete. It is proposed that multi-physical hydration models be linked to microstructure characterization and transport properties in order to examine ITZ microstructure development.

The additional of SAP could also enhance the degree of hydration and densify the cement matrix on the surrounding SAP [192,193]. However, the water release from SAP can leaves pores [110,194] and undermine the properties of the cement matrix [181]. Studies revealed by Jianhui [195] reported that the development of pores can impair the ITZ and bond strength between SAP and the cement matrix as shown in Figure 12. This effect increased as the SAP content and particle size of the SAP increased. Ridi et al. [196] found that the incorporation of SAP into high-performance concrete (HPC) helps the densification of the microstructure of the concrete. They also discovered via experimentation that the self-curing procedure is associated with increased strength, a higher degree of hydration, and penetration resistance in heat-cured concrete, as well as a better microstructure with a lower porosity in HPC.

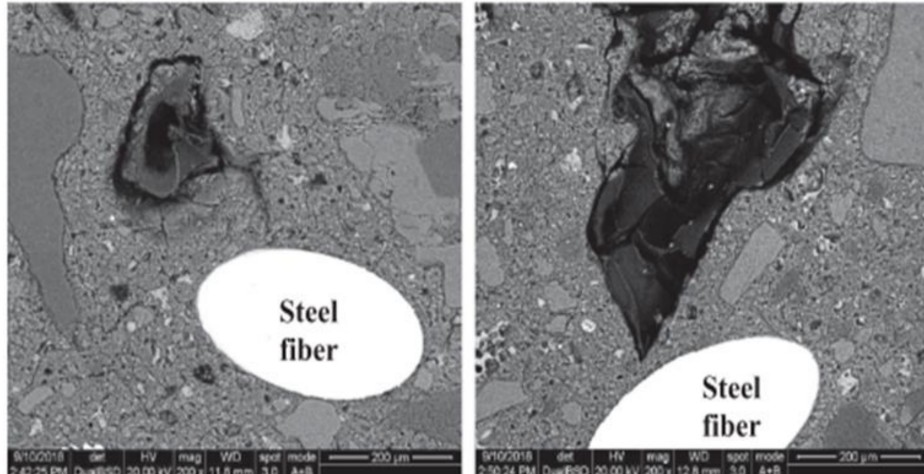

**Figure 12.** The microstructure of the high-performance concrete with SAP [195].

## 5. Environmental Benefits of Self-Curing Concrete or Mortar

The utilization of natural aggregates has become a growing issue, given the excessive use for construction purposes. In addition, to reduce wastage and recycling, by-products have attracted growing interest and attention from researchers. As such, the development of new methods in managing waste has swiftly become one of the most important research topics. The increasing need to reuse materials, given diminishing natural resources, has become a recognized and volatile issue healthy debated by scholars and researchers. Accordingly, research studies have increasingly investigated artificial aggregates for infrastructure construction as a replacement for natural aggregates. Most artificial aggregates produced from waste or by-products, [42,197–199] and their unique properties, such as porosity and absorption capacity, have the added benefits of supplying water internally for self-curing concrete [48,84,163]. In addition, using large quantities of industrial and agricultural waste can help to reduce suspension in natural aggregates [200,201]. Moreover, it is seen as an effective process to convert waste or by-products into valuable building materials. Thus, this leads to beneficial performance for the economy whereby it produces cheaper mortar and concrete materials for low-cost construction [42]. The utilization of artificial aggregates produced from waste materials will also aid in the reduction and usage of materials made from chemical resources as curing agents in concrete. Furthermore, it positively contributes to safeguarding and protecting the environment leading to sustainable development and reduces carbon emission [200]. Besides, waste material as a curing agent replacing natural aggregates in concrete will lead to the reduction in landfills and prevent natural resources such as flora and fauna from being destroyed [199].

The review of research on self-curing concrete revealed it to be both dense and durable compared to conventional non-cured concrete, [124,138,202] thus increasing the building's lifespan (i.e., service life). Therefore, demolition activities and maintenance work on such structures can reduce concrete rubble as waste construction material. The growing scarcity of water resources in hot climates, such as in Afro-Asian regions, has also required regular checking on the use of freshwater for concrete production, given the approximate rate of 3 m$^3$ for every 1 m$^3$ of concrete produced for curing purposes [28,56]. Therefore, the techniques for self-curing concrete need to be monitored and advanced, where possible, by conserving the use of freshwater. For example, unneeded water spraying or water sprinkling techniques are needless and should be avoided. Therefore, the self-curing curing agent provides an avenue for further cement hydration via the absorption of water before or during the mixing of concrete as the absorbed water can be slowly released during the process of hardened concrete [32,138].

**Table 2.** Effect of self-curing agents and curing condition on concrete workability, engineering, and microstructure performance.

| Curing Agent | Concrete Type | Curing Agent Replacement | Curing Condition | Finding | Ref. |
|---|---|---|---|---|---|
| Lightweight aggregate Leca/Expanded shale | Normal-strength concrete | Leca 12.5–50% Shale 10–30% | Saturated-surface dry (SSD)/water | Workability: The use of lightweight aggregates did not alter the predesigned slump of the mix due to the incorporation of the aggregate in a pre-soaked saturated surface dry condition. Strength at 28 days: For Leca: Reduced by 22% to 29%. For Shale: Reduced by 8% to 18%. The lowered compressive strength did not violate the minimum strength required for highway construction projects of 3000 psi. Reduction ratio of autogenous shrinkage at 28 days (%): For Leca: Reduced by 7.5% to 25%. For Shale: Reduced by 10% to 25%. | [38] |
| Lightweight aggregate Leca/Fly ash | High-performance concrete | For both 5–25% | Saturated-surface dry (SSD) air | Workability: Increased by 0.5 to 1% for both Leca and Fly ash. Slump flow of all the mixes with Leca and fly ash varied from 680 to 710 mm which is within the desired range of 650 to 800 mm. | [203] |
| Lightweight aggregate Volcanic tuff | High-performance concrete | 5–20% | Saturated-surface dry (SSD)/water | Workability: Slump values reduced with an increase in the LWA content. Even LWA prepared in SSD condition, assuming that LWAs could have lost some amount of internal water giving the pores free space to absorb additional water from the mixture. Strength at 28 days: Reduced by 2.3% to 18.6%. Reduction ratio of autogenous shrinkage at 28 days (%): 33% to 54%. | [204] |

Table 2. *Cont.*

| Curing Agent | Concrete Type | Curing Agent Replacement | Curing Condition | Finding | Ref. |
|---|---|---|---|---|---|
| Lightweight aggregate Autoclaved Aerated Concrete aggregate (AAC) | High-performance concrete | 20–60% | Saturated-surface dry (SSD)/water/air | Slump: The slump value was increased due to water retain ability, extra water on the surface of AAC particles. Compared to control sample (580 mm), the slump value increased to 890, 950, and 1060 mm with increasing LWAs to 20%, 40%, and 60%, respectively.<br>Strength at 28 days:<br>Water curing: Reduced by 11.4% to 36%.<br>Air curing: For 20 and 40% achieved increment by 0.2 to 5.4%. However, the increasing level of LWAs to 60% led to drop the strength by 25%.<br>Microstructure: Reserved water in AAC aggregates would, be transferred to cement paste across ITZ, increasing hydration level to the cement binders. The strength improvement in later age was mainly influenced by more C-S-H formation and denser microstructures. The usage of AAC-LWA in SSD condition would provide higher strength in all cases than the as- received/dry AAC-LWA. | [104] |
| Lightweight aggregate Leca | High-performance concrete | 10–25% | Saturated-surface dry (SSD)/water curing, followed by air curing | Strength at 28 days: With increasing LWAs content, the gain in strength dropped from 15% to 2%. | [205] |
| Lightweight aggregate Leca/PEG | Normal-strength concrete | Leca 10–20% PEG 1–3% | Saturated-surface dry (SSD)/air | Strength at 28 days:<br>For Leca: Increased by 5 to 13.3%.<br>For PEG: Increased by 13.3 to 15%.<br>Compressive strength systematically increases as leca increased. This may be attributed to the continuation of the hydration process by store water in the SSD Leca resulting in lower voids and pores and greater bond force between the cement paste and aggregate. | [79] |
| Polymer/PEG | Normal-strength concrete | Polymer (SAP) 0.35–1% PEG 0.25–2% | Air | Strength at 28 days:<br>For polymer: The gain on strength trend to increase with increasing SAP content and reduction in water/cement ratio from 0.60 to 0.40%.<br>For PEG: Similar trend of results was observed and the gain in strength trend to increase with increasing PEG content and reduce the water/cement ratio. | [56] |

**Table 2.** *Cont.*

| Curing Agent | Concrete Type | Curing Agent Replacement | Curing Condition | Finding | Ref. |
|---|---|---|---|---|---|
| Polymer | High-performance concrete | 0.2–0.6% | Saturated-surface dry (SSD) | Strength: Reduced by 3.3% to 10%. The strength loss was more accentuated when higher contents of SAP were used regardless of size and type of SAP. The negative effect of SAP on strength development of UHPC was related to the formation of voids after the release of the absorbed water from SAP to the matrix. Shrinkage: Reduced by 33% to 66.7%. The increase in the content of SAP, regardless of size, reduced the autogenous shrinkage significantly. | [156] |
| Polymer | High-performance concrete | 0.3% | Air-dry (AD) | Workability: Enhance the workability performance. Strength: Reduced the compressive strength by 8.5%. It may be concluded that the shape and size of the SAP particles may have a major influence on the strength values. Shrinkage: Reduced by 50%. Polymer mitigates the autogenous shrinkage ofconcrete very effectively | [110] |
| Polymer | High-performance concrete | 0.17–0.49% | Saturated-surface dry (SSD)/air | The HPC internally cured with SAPs showed a lower autogenous shrinkage than that without SAPs, which was due to the decreased self-desiccation of HPC internally cured with SAPs. | [157] |
| PEG/Polymer | High-performance concrete | PEG 0.5–2.0% SAP 0.25–1.0% | Saturated-surface dry (SSD) | Strength at 28 days: For PEG: Reduced by 0.4 to 8%. For polymer: Reduced by 7.6 to 24.6%. | [34] |
| PEG | Normal strength concrete/High-performance concrete | 0.1–1.0% | Water/Air/Sealed | Workability: Improved with inclusion of PEG. With increase in dosage of PEG, the flowability of concrete has improved. This is due to low viscous nature.Strength at 28 days: Enhanced by 14.3 to 25%. Microstructure: Use of PEG chemicals resulted in a dense microstructure with a calcium/silica of 1.12. This is an indication of stable C-S-H gel formation compared to non-cured specimens. While non-cured specimens exhibited poor microstructure with interlinking of micro cracks and ettringite formation. Test involved XRD and SEM. | [55] |

**Table 2.** *Cont.*

| Curing Agent | Concrete Type | Curing Agent Replacement | Curing Condition | Finding | Ref. |
|---|---|---|---|---|---|
| Natural fibres | High-performance concrete | 0.8–1% | Saturated-surface dry (SSD)/Air-dry (AD) | Microstructure: Internal curing mechanisms of hardwood eucalyptus pulp-cement composites and then uses SEM, XPS, and AFM to examine interactions between the fiber and surrounding hydrating cement. Addition of eucalyptus pulps as internal curing agents may slightly delay setting. Observations by XPS show development of calcium silicate hydrate and calcium hydroxide at cement paste. | [33] |
| Natural fibres | High-performance concrete | 1.2–2.4% | Air-dry (AD) | Tests included in microstructure test are SEM, Atomic Force Microscopy (AFM), and Nanoindentation test in order to examine the effect of kenaf on the hydration and micromechanics of concrete. | [58] |
| Polymer/Micritic calcite aggregate (ANWA) | High-performance concrete | 100% | Saturated-surface dry (SSD)/Air-dry (AD)/Sealed | Workability: the workability of concrete enhanced with inclusion both polymer and micritic calcite aggregates. The concrete mixture achieved a slight increased slump that was attributed to the instant adhesion of cement particles on the aggregates surface that smoothed out their rough surface. Strength at 28 days: For both curing condition and curing agents, the compressive strength reduced by 0.9 to 19%. Shrinkage at 28 days: Reduced by 42.9 to 96.4%. Concrete mixtures containing self-curing agent exhibited a significant reduction of autogenous shrinkage. | [206] |
| Porous ceramic aggregate (PCA) | High-performance concrete | 10–20% | Saturated-surface dry (SSD) | Strength at 28 days: The compressive strength of concrete with PCA is higher than control concrete. In particular, the compressive strength of 20% is more than 15% higher than control concrete at age 28 days. Microstructures: Consequently, this study shows the possibility that the internal curing water supply improves the micro-hardness of the ITZ around PCA. The improvement of the ITZ leads to an increase in the compressive strength of concrete | [48] |

**Table 2.** *Cont.*

| Curing Agent | Concrete Type | Curing Agent Replacement | Curing Condition | Finding | Ref. |
|---|---|---|---|---|---|
| Porous ceramic aggregate (PCA) | Normal strength concrete. | 10–20% | Saturated-surface dry (SSD) | Workability: Enhancement on slump values were observed. Strength at 28 days: A 10% replacement of coarse aggregate by PCA was more effective in improving compressive strength than a 20% replacement by PCA at the early ages of 3 and 7 days, independent of exposure conditions. Shrinkage at 28 days: Internal curing using PCA to replace part of the coarse aggregate was not effective in reducing autogenous shrinkage, which could be explained by the comparatively high water-to-cement ratio of 0.55 of the present concrete. | [100] |

## 6. Conclusions

After undertaking the literature review in this field, it can be concluded that many researchers have investigated the influence of different materials as curing agents toward physical and mechanical properties, shrinkage (autogenous and drying), and microscopic in self-curing concrete. Evidence supports the fact that all curing agents can be used given their capability to absorb water and act as a water reservoir in concrete, thereby gradually releasing water for the hydration process. The use of curing agents was also shown to have a favorable effect on the properties of self-curing concrete, comparable to conventional concrete.

The effect of self-curing by porous aggregate and SAP depends on the amount, particle size, and pore structure of curing agent in concrete. If the amount, particle size, and pore structure of those curing agents substantially increase, it may cause a substantial decrease in strength in high-performance cement-based materials. Many studies have reported an increase in compressive strength of high-performance concrete when the optimum proportion of replacement of conventional aggregate is obtained. The incorporation of internal curing water would reduce total shrinkage at early and later age by change in internal RH and the saturation degree of capillary pores. As a result, this enhances hydration and refines pore structures in cementitious materials. A self-curing agent promotes the hydration of ITZ and improves the bonding between cement paste matrix and porous aggregate or SAP or PEG, thus increasing the density of ITZ. This improvement reduces porosity and pore connectivity, leads to reduced permeability, and finally enhances the strength and durability of cementitious material.

Based on the work and requirements that have been highlighted in this study, a number of recommendations are purposed for future research: (i) The curing effect is difficult to control since typical curing materials are only capable of absorbing and desorbing the water and the uniform distribution in concrete makes it difficult to have artificial modification. Hence, intelligent materials are introduced, such as humidity sensors embedded in microcapsules; this may improve internal curing conditions. (ii) There are many investigations on the effect of internal curing of high-performance cement-based materials. However, the studies on the effect of normal strength of self-curing cementitious material is very limited. Therefore, the investigation on the physical, mechanical, and microscopic properties of normal strength of self-curing cementitious materials is looked into. (iii) Particle size, amount, and distribution in concrete of porous aggregate and self-curing agent SAP, as well as the amount of internal curing water, only exist in theoretical analysis. There is no suitable or acceptable technique to define the ITZ range between porous aggregate or SAP and cement paste. Still, the specified method is significant for designing the amount, size, and distribution of a porous aggregate or SAP and internal curing water.

**Author Contributions:** N.H.: Conceptualization, Methodology, Writing—Original draft preparation; H.M.S.: Validation, Supervision; M.H.B.: Verified the manuscript structure and supervised the overall research. A.R.M.S.: Writing—review & editing, Supervision; M.N.M.S.: Visualization, Validation; I.F.: Visualization, Validation; O.B.: Visualization, Validation; G.F.H.: Project administration, Supervision, Conceptualization, and Methodology. All authors have read and agreed to the published version of the manuscript.

**Funding:** This research was supported by funding from the Department of Manufacturing and Civil Engineering, Norwegian University of Science and Technology (NTNU). The authors extend their appreciation to Researchers Supporting Project numbers (Q.J130000.2409.04G49) and (Q.J130000.2409.04G50), Universiti Teknologi Malaysia.

**Institutional Review Board Statement:** Not applicable.

**Informed Consent Statement:** Not applicable.

**Data Availability Statement:** Data is contained within the article.

**Acknowledgments:** The authors would like to thank Universiti Teknologi MARA (UiTM) and Universiti Teknologi Malaysia (UTM), the Ministry of Higher Education, and laboratory staff for their support while completing this paper.

**Conflicts of Interest:** The authors declare no conflict of interest.

## Abbreviations

| | |
|---|---|
| LWA: | Artificial lightweight aggregate |
| SAP: | Superabsorbent polymers |
| PEG: | Polyethylene glycol |
| ANWA: | Artificial normal-weight aggregate |
| LECA: | Expanded clay |
| RH: | Rice husk ash |
| ITZ: | Interfacial transition zone |
| C-S-H: | Calcium silicate hydrate |
| OPC: | Ordinary Portland cement |
| CH: | Calcium hydroxide |
| $Ca_3SiO_5$: | Alite |
| $Ca_2SiO_4$: | Belite |
| $Ca_4Al_2Fe_2O_{10}$: | Ferrite |
| $Ca_3Al_2(OH)_{12}$: | Hydrogarnet |
| $Ca_3Al_2O_6$: | Aluminate |
| $CaSO_4 \cdot 2H_2O$: | Gypsum |
| $[Ca_3Al(OH)_6 \cdot 12H_2O]_2(SO_4) \cdot 32H_2O$: | Ettringite |
| $[Ca_2Al(OH)_6 \cdot 2H_2O]_2(SO_4) \cdot 2H_2O$: | Monosulfate |
| HPC: | High-performance concrete |
| SCMs: | Supplementary cementitious material |

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
