# Peer review of "A Review on the Use of Self-Curing Agents and Its Mechanism in High-Performance Cementitious Materials"

_buildings, doi:10.3390/buildings12020152_

Round 1
Reviewer 1 Report
This paper presents a comprehensive review of the use of self-curing agents in high-performance cementitious materials. This paper can be accepted for publication.
Author Response
As attached

Reviewer 2 Report
It was expected that this review paper will stand alone as a paper that saves a reader from reading hundreds of papers in the field of self-curing agents and related mechanisms. Certainly, I have no technical comments as this is a review paper. However, this paper reports a few previous studies about the effect of self-curing agents on the properties of concrete/mortar, which is hard to follow. All the details which are compiled in sections 3.1, 3.2, 3.3 and 3.4 should be also reported in the form of Table. To enhance the novelty (contributions to the relevant field) I suggest providing more detailed results and outcomes in the form table.
Concrete and mortar are two different forms of composites, hence should be discussed separately and proper differences should be outlined. Very frequently these two parameters are used together!
Conclusions and future recommendations can be written as one section as there are only a few things authors had to conclude.
Author Response
As attached

Reviewer 3 Report
The article entitled A review on the use of self-curing agents and its mechanism in high-performance cementitious materials is very interesting. The article is of a review nature. No new information is brought in, but a very large number of papers on the subject under review are skillfully reviewed. I found chapter 3 interesting, summarizing and comparing various self-curing techniques for concretes and mortars.
I believe that the subject matter of the article does not quite fit the scope of the journal. The topics presented are definitely better suited to, for example, the journal Materials.
Editorial Notes:
â–ª line 131 and 134 - error in writing the unit - it is g/cm3 and should be g/cm3
â–ª line 368 - mistake in writing the unit - it is m3 and should be m3
â–ª incomplete (usually no publisher) literature references - items: 1, 22, 25, 35, 36, 37, 59, 103, 141, 151, 188,
â–ª in the literature list in item 182 the number [182] was unnecessarily repeated
Author Response
As attached

Round 2
Reviewer 2 Report
It can be accepted. Congratulations.